# Str-GCL: Structural Commonsense Driven Graph Contrastive Learning

## Abstract

Graph Contrastive Learning (GCL) is a widely adopted approach in unsupervised representation learning, utilizing representational constraints to derive effective embeddings. However, current GCL methods primarily focus on capturing implicit semantic relationships, often overlooking the structural commonsense embedded within the graph's structure and attributes. This structural commonsense is crucial for effective representation learning. Identifying and integrating such structural commonsense in GCL poses a significant challenge. To address this gap, we propose a novel framework called Structural Commonsense Unveiling in Graph Contrastive Learning (Str-GCL). Str-GCL leverages first-order symbolic logic rules to represent structural commonsense and explicitly integrates these rules into the GCL framework. Specifically, we introduce structural commonsense from both topological and attribute rule perspectives, processing these rules independently without modifying the original graph. Additionally, we design a representation alignment mechanism that guides the encoder to effectively capture this structural commonsense. To the best of our knowledge, this is the first attempt to directly incorporate structural commonsense into GCL in a rule-based manner. Extensive experiments demonstrate that Str-GCL significantly outperforms existing GCL methods, providing a new perspective on leveraging structural commonsense in graph representation learning.

## CCS Concepts

• **Mathematics of computing → Graph algorithms**.

## Keywords

Graph Neural Networks, Graph Contrastive Learning, Structural Commonsense

**ACM Reference Format:**
Anonymous Author(s). 2018. Str-GCL: Structural Commonsense Driven Graph Contrastive Learning . In *Proceedings of Make sure to enter the correct conference title from your rights confirmation emai (Conference acronym 'XX).* ACM, New York, NY, USA, 13 pages. https://doi.org/XXXXXXX.XXXXXXX

## 1 Introduction

Graph Representation Learning (GRL) has emerged as a powerful strategy for analyzing graph-structured data over the past few

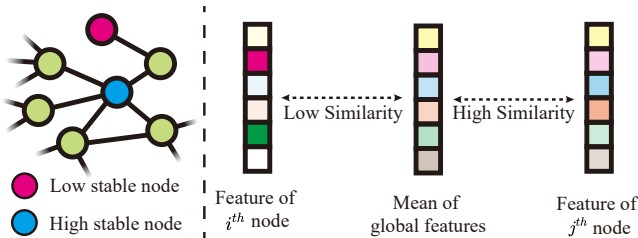

**Figure 1: Two basic structural commonsense in graph data based on topology and attributes. Topologically, nodes with high-degree neighbors exhibit high stability during training, while nodes with fewer neighbors show lower stability. In terms of attributes, $i^{th}$ node possesses features that are significantly aligned with its label, making it easy to classify, whereas $j^{th}$ node has features that are too similar to the globals, rendering its label-aligned features ambiguous.**

years. By using Graph Neural Networks (GNNs) [1], GRL has garnered significant attention, aiming to transform nodes into effective low-dimensional embeddings. However, most GNN models train under supervised or semi-supervised scenarios, which requires a large number of labels. These methods are intricate and expensive in a growing explosion of graph-structured data. In contrast, graph self-supervised learning (GSSL), such as the representative Graph Contrastive Learning (GCL) methods [2, 3], does not require labels to acquire node embeddings. These methods have achieved performance comparable to their supervised counterparts for most graph representation learning tasks, such as node classification [4–6], graph classification [7–9] etc.

Existing GCL methods [3, 10, 11] commonly utilize the InfoNCE principle to generate effective node representation, which encourages the model to maximize the similarities between positive samples and minimize the similarities between negative samples during training. These samples are typically established through two views generated by graph augmentations, such as edge removal and feature masking. Some researchers refine the optimization strategy of InfoNCE by exploring various strategies, such as leveraging negative samples [11] or considering graph homophily [12]. Additionally, some approaches [13–15] employ two independent encoders, with one encoder designed to learn the node representation from the other. Furthermore, some studies [16–18] explore GCL from the perspective of homophily and heterophily.

Despite the significant advancements in GCL, current GCL methods often operate as black boxes with limited explainability, making it difficult to understand or trust their decision-making processes and fully assess their learning capabilities. Our observations reveal that a significant proportion of nodes are consistently misclassified across multiple experiments with existing GCL models, such as GRACE [3] (as detailed in Section 2). Relying solely on learning

implicit relationships proves insufficient for adequately training the encoder, preventing the model from capturing more complex or nuanced patterns of the graph. This limitation represents a fundamental performance bottleneck that existing methods cannot address. Through a detailed analysis of these misclassified nodes, we find that many of them can be correctly classified only with the aid of expert knowledge, which led us to a key question: Could there be structural commonsense embedded within graph structures that we are overlooking? Furthermore, could we develop an interpretable GCL approach that explicitly incorporates structural commonsense to improve both model performance and interpretability?

However, integrating these intuitive structural commonsense into GCL models presents significant challenges. First, how can we discover these intuitive structural commonsense? Unlike knowledge graphs [19, 20], which contain abundant triples that offer clear guidance, general graph data lacks such explicit information. In an unsupervised setting without labels, these rules are even harder to detect and interpret. Second, how can we represent and incorporate them into the model? Even if we manage to identify these intuitive structural commonsense, effectively encoding them and enabling GCL models to recognize and leverage them appropriately remains a complex technical obstacle.

To address these challenges, we propose a novel GCL model called **Str**uctural commonsense Driven **G**raph **C**ontrastive **L**earning (**Str-GCL**), which explicitly integrates structural commonsense into the learning process to enhance effectiveness and interpretability. Specifically, we introduce structural commonsense from both topological and attribute perspectives (as illustrated in Figure 1), formulating two representative basic rules expressed using first-order logic. Even in unsupervised settings without labels, these rules can capture structural patterns that are intuitively perceptible to humans. Furthermore, Str-GCL independently generates rule-based representations and employs a representation alignment mechanism to effectively integrate these rule-based and node-based representations. By embedding structural commonsense into the model using first-order logic rules, our approach enables the encoder to perceive and leverage additional structural knowledge, allowing it to focus on more intricate and nuanced patterns within the graph. This integration ultimately enhances both the model's performance and its interpretability.

Our main contributions are summarized as follows:

- We are the first to pose the problem of integrating structural commonsense into contrastive learning, which primarily involves how to leverage human intuition to uncover structural commonsense present in graph data (knowledge that is often overlooked by traditional GCL methods) and how to effectively encode this commonsense to enable GCL models to recognize and utilize it.
- We propose a novel graph contrastive learning paradigm, called Str-GCL, that uses first-order logic to express rules and guides the model to learn structural commonsense. To the best of our knowledge, this is the first attempt that human-defined rules are explicitly introduced into GCL, providing an interpretable approach from the perspective of structural commonsense.

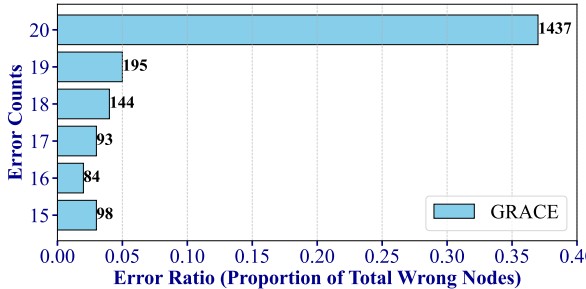

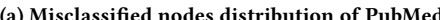

(a) Misclassified nodes distribution of PubMed

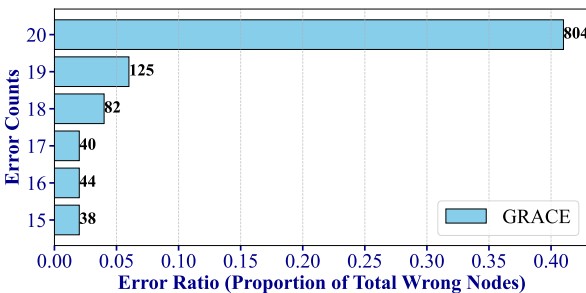

(b) Misclassified nodes distribution of CS

Figure 2: Misclassified nodes distribution of PubMed and CS datasets. The Error Ratio (horizontal axis) represents the percentage of nodes misclassified a specific number of times relative to the total number of misclassified nodes, while the Error Counts (vertical axis) represents the number of times a node is misclassified across 20 independent tests. To illustrate that some nodes frequently exhibit classification errors, we include only those nodes that are misclassified 15 or more times.

- We conduct experiments on six datasets, evaluating our model's performance by comparing it with numerous other GCL models in classification and clustering tasks. Additionally, we perform detailed data analysis on misclassified nodes and compare our results with the baseline model. Moreover, we integrate Str-GCL as a plugin into multiple GCL baselines, enhancing their performance to verify the extensibility of Str-GCL. Extensive experiments and visualization demonstrate the effectiveness of Str-GCL.

## 2 Observation & Analysis

In this section, we aim to detect nodes that are not adequately learned by the model, as manifested by their frequent misclassification in multiple tests across several benchmark datasets [21]. Here, we use PubMed and CS as the representative examples. Specifically, for each dataset, we run the well-known GCL method GRACE [3] 20 times under the default experimental settings. As shown in Figure 2, we observe that approximately 40% of the misclassified nodes are consistently misclassified across all training runs, indicating that a significant portion of nodes are not adequately constrained by the objective function during training. Therefore, we analyze

 

the attributes and topological properties of the misclassified nodes (those with error counts greater than or equal to 15). We attempt to manually classify these nodes based on their connections and feature similarity while masking their labels. We find that by considering only connectivity and similarity, we can manually identify that many misclassified nodes and their neighbors belong to the same class (as illustrated by the simple example in Figure 1). However, the trained GCL model fails to recognize these misclassified nodes. This leads us to understand that, even though humans can easily interpret such simple structural commonsense, the current GCL paradigm is incapable of perceiving or learning them. Instead, GCL focuses on constraining instances in the representation space, overlooking the inherent general structural commonsense in the topology of data. This observation inspires us to explore structural commonsense within the distribution of error-prone nodes and to devise targeted interventions to mitigate their misclassification.

## 3 Preliminaries

**Notations** Given a graph $\mathcal{G} = (\mathcal{V}, \mathcal{E})$, where $\mathcal{V} = \{v_1, v_2, \cdots, v_N\}$ is the set of nodes, $\mathcal{E} \subseteq \mathcal{V} \times \mathcal{V}$ is the set of edges. Additionally, $X \in \mathbb{R}^{N \times F}$ is the feature matrix, and $A \in \{0, 1\}^{N \times N}$ is the adjacency matrix. $X_i \in \mathbb{R}^F$ is the feature of $v_i$, and $A_{ij} = 1$ iff $(v_i, v_j) \in \mathcal{E}$. Our objective is to learn an encoder $f(X, A) \in \mathbb{R}^{N \times F'}$ to represent high-level representations under the unsupervised scenarios, which can be used in various downstream tasks.

**Graph Contrastive Learning (GCL)** To illustrate our approach, we employ a classic GCL method, GRACE [3], as a case study. Giving a graph $\mathcal{G}$, two augmentation functions $t_1$ and $t_2$ are applied to the original data, resulting in two augmented views $t_1(\mathcal{G}) = \mathcal{G}_1 = (X_1, A_1)$ and $t_2(\mathcal{G}) = \mathcal{G}_2 = (X_2, A_2)$. Subsequently, these augmented views are processed by a shared GNN encoder, and then generate node representations $U = f(X_1, A_1)$ and $V = f(X_2, A_2)$. Finally, the loss function is defined by the InfoNCE [22] loss as:

$$\ell(u_i, v_i) = \log \frac{e^{\theta(u_i, v_i)/\tau}}{e^{\theta(u_i, v_i)/\tau} + \sum_{k \neq i} e^{\theta(u_i, v_k)/\tau} + \sum_{k \neq i} e^{\theta(u_i, u_k)/\tau}},$$ (1)

where $\theta(\cdot, \cdot)$ is the cosine similarity function and $\tau$ is a temperature parameter. The positive samples are the node pairs $(u_i, v_i)$, representing corresponding nodes in two views, and the negative samples are other node pairs $(u_i, v_i)$ and $(u_i, u_k)$ where $k \neq i$. Since two graph views are symmetric, $\mathcal{L}_{\text{InfoNCE}}$ can be given by:

$$\mathcal{L}_{\text{InfoNCE}} = \frac{1}{2N} \sum_{i=1}^{N} (\ell(u_i, v_i) + \ell(v_i, u_i)).$$ (2)

There are many types of GNN now, which can be served as the encoder. We use a graph convolutional network (GCN) [23] as our encoder $f$ by default, which can be formalized as $f(X, A) = H = \hat{A}XW$, where $\hat{A} = \tilde{D}^{-1/2}(A + I_N)\tilde{D}^{-1/2}$. Here, $\hat{D}$ represents the degree matrix of $A + I_N$, and $I_N$ represents the identity matrix. $W$ represents learnable weight matrix.

**First-Order Logics (FOLs)** FOLs is a formal system used to represent relationships and properties through predicates, quantifiers (including the universal quantifier ($\forall$) and the existential quantifier ($\exists$)), and logical connectives (including conjunction ($\wedge$),

disjunction ($\vee$), negation ($\neg$) and implication ($\rightarrow$)). In the context of graph data, first-order logic enables precise description and manipulation of node and edge relationships. For example, the predicate Connected($v_i, v_j$) can denote an edge between $v_i$ and $v_j$, while HasFeature($v_i$, feature) indicate that $v_i$ has a specified feature.

## 4 Str-GCL

In this section, we explore embedding general structural commonsense set by humans into models in the form of rules, and analyze these rules in various datasets, with a special emphasis on homogeneous graphs. We detail the specific implementation aspects of Str-GCL, providing a comprehensive understanding of its framework. The model architecture is illustrated in Figure 3.

### 4.1 General Structural Commonsense Expressed by Symbolic Logic

To uncover patterns not readily discernible within GNNs, and to aid the training of encoders based on these patterns, our model incorporates general structural commonsense derived from human intuition. Through observation and statistical analysis, we have identified that sets of nodes adhering to certain observable human patterns are more prone to misclassification compared to those outside these patterns. These statistically derived intuitions serve as a bridge between error-prone nodes and observed patterns. Moreover, these commonsense insights are represented using the expressive power of first-order logic.

**Neighborhood Topological Summation Constraint (NTSC)** NTSC operates on the premise that the attributes of a node's neighbors can significantly influence the representations it generates after passing through the encoder. In this paper, we use GCN as the encoder, the first-order neighbors have the greatest impact on the node. Specifically, this rule targets nodes with limited topological connections, assigning higher attention to nodes with lower aggregate neighbor degrees. The underlying hypothesis is that a low sum of first-order neighbor degrees may not be able to effectively learn the local graph structure and lack reliable information for generating effective representations. Formally, we can represent NTSC using first-order logic as follows:

$$\forall v_i \forall v_j (\text{Neighbor}(v_i, v_j) \rightarrow v_j \in \mathcal{N}(v_i)),$$
$$\forall v_i (\text{TotalDegree}(v_i) = \sum_{v_j: Neighbor(v_i, v_j)} \deg(v_j)),$$ (3)

where $\mathcal{N}(v_i)$ represents the set of first-order neighbors of node $v_i$, Neighbor($v_i, v_j$) is a function which represents the total sum of degrees of all neighbors of $v_i$, $\deg(v_j)$ is a function that returns the degrees of node $v_j$. We use $d$ to represent the degree of a node, and $d_{\text{sum}}$ represents the sum of the degrees of each node's neighbors, i.e., $d_{\text{sum}} = A \cdot d$. To avoid the excessive influence of large differences in node degrees, we perform logarithmic normalization on $d_{\text{sum}}$ as $\hat{d}_{\text{sum}} = \log(1 + d_{\text{sum}})$. Finally, we normalize values to generate weights: $w_i = \max(\hat{d}_{\text{log}}) - \hat{d}_{\text{log}}(i)$. In this way, smaller degrees will be assigned larger weights, thus paying more balanced attention to nodes with different structures during the learning process.

**Local-Global Threshold Constraint (LGTC)** In the node classification task on homophilic graphs, nodes that exhibit substantial disparities between their neighbor's feature similarity and the

global feature similarity may have more unique features or more distinct structures. On the contrary, when the neighbor's feature similarity is strikingly similar to the global feature similarity, it may indicate that the node's features or local structure are unclear or unspecific. Therefore, the goal of LGTC is to measure the gap between the two similarities. The underlying assumption is that a low value may lack unique information about their own class. Formally, we can represent LGTC using first-order logic as follows:

$$\forall v_i (\mathrm{LocalSim}(v_i) = \mathrm{avg}_{v_j \in \mathcal{N}(v_i)} \mathrm{sim}(v_i, v_j)),$$
$$\forall v_i (\mathrm{GlobalSim}(v_i) = \mathrm{avg}_{v_j \in \mathcal{G}} \mathrm{sim}(v_i, v_j)), \quad (4)$$

where $\mathrm{LocalSim}(v_i)$ is a function representing the average similarity of node $v_i$ to its first-order neighbors $\mathcal{N}(v_i)$, $\mathrm{GlobalSim}(v_i)$ is a function representing the average similarity of node $v_i$ to all other nodes in $\mathcal{G}$. Specially, we apply Principal Component Analysis (PCA) [24] to the $X$ to capture the most significant invariance: $X' = \mathrm{PCA}(X)$. We calculate the average similarity $\mathrm{AS}(v_i)$ between node $v_i$ and its neighbor $\mathcal{N}(v_i)$ using cosine similarity, and obtains a global similarity $\mathrm{GS}(v_i)$: $\mathrm{AS}(v_i) = \frac{1}{|\mathcal{N}(v_i)|} \sum_{v_j \in \mathcal{N}(v_i)} \mathrm{sim}(X'_i, X'_j)$, $\mathrm{GS}(v_i) = \frac{1}{|N|-1} \sum_{v_j \in N \setminus v_i} \mathrm{sim}(X'_i, X'_j)$. Then, we compute the normalized difference $\mathrm{Diff}(v_i)$ between $\mathrm{AS}(v_i)$ and $\mathrm{GS}(v_i)$: $\mathrm{Diff}(v_i) = \frac{1}{2}(\mathrm{AS}(v_i) - \mathrm{GS}(v_i) + 1)$. Finally, we generate similarity-based weights $s_i$: $s_i = \max_{v_j \in N} \mathrm{Diff}(v_j) - \mathrm{Diff}(v_i)$. By subtracting each node's normalized difference from the maximum, higher attention is given to nodes with smaller differences, ensuring those closer to global features receive more balanced attention during learning.

After NTSC and LGTC, we use $\mathrm{MLP}_{\mathrm{param}}$ to learn its weights, i.e., $q = \sigma\left(\mathrm{MLP}_{\mathrm{param}}([w; s])\right)$. The rule representations are generated by an independent MLP acting on the features $W$. The generated weights act on the rule representations to generate the final $H_R$ for subsequent $\mathcal{L}_{\mathrm{cross}}$ alignment.

## 4.2 Loss Function Design

As demonstrated, NTSC and LGTC significantly contribute to the overall performance of the GCL model. Motivated by these findings, we propose a targeted strategy to extract and individually train features of nodes identified by these rules. Eventually, we will generate a two-part representation, a rule representation and a complete node representations generated by the encoder. We then designed a representation alignment mechanism that employs a specific loss function to constrain these two representations. This ensures that the node representations can perceive the defined structural commonsense implicit in the rule representations. Ideally, if rule-based representations are directly applicable to downstream tasks, all nodes identified by these rules as error-prone will be correctly classified. However, due to the inconsistency between the rule-based embedding space and the full node representations, and the unpredictable nature of logical relationships in graph data, this scenario is rarely achievable. Directly applying rule representations to downstream tasks often neglects critical information, as these representations fail to capture the complete graph topology and the most suitable embedding space. Additionally, the quality of node representations may significantly degrade due to noise introduced by directly incorporating rule representations, which integrate entirely different distribution representations into the node's own.

To address this challenge, we first use a separate contrastive loss for the rule representations, so that similar samples are closer while dissimilar samples are further away from each other in the rule representations. Details are as follows:

$$\mathcal{L}_{\mathrm{rule}} = -\frac{1}{N} \sum_{i=1}^{N} \log\left(\frac{S_{ii}}{\sum_{j=1}^{N} S_{ij}}\right), \quad (5)$$

where $S = f(Z_{\mathrm{norm}} Z_{\mathrm{norm}}^T)$, $Z_{\mathrm{norm}}$ is the normalized rule representations, $S_{ij}$ represents the similarity of rule representations between $v_i$ and $v_j$, and $f(x) = e^{(x/\tau)}$.

After processing of rule representations, we use a representation alignment mechanism to design the loss between these representations. Specifically, we avoid the problem of introducing noise by aligning the distribution of rule representations and original representations. We also separate rule representations and enable the rule representations to perceive the information of the node representations. Now, we have node representations $H_N$ and rule representations $H_R$, respectively, with dimension $n \times d$ where $n$ is the number of error-prone nodes and $d$ is the number of representations. The mean of these representations is computed as:

$$\mu_N = \frac{1}{n} \sum_{i=1}^{n} H_{N_{i,:}}, \quad \mu_R = \frac{1}{n} \sum_{i=1}^{n} H_{R_{i,:}}, \quad (6)$$

where $H_{N_{i,:}}$ and $H_{R_{i,:}}$ are the rows of the representations $H_N$ and $H_R$. These means provide a central node around which the representations are distributed. Then we compute the covariance matrix, which measures how much two random variables change together and indicates the spread and orientation of the data distribution:

$$\mathrm{Cov}(H_N) = \frac{1}{n-1}(H_N - \mu_N)^\top (H_N - \mu_N),$$
$$\mathrm{Cov}(H_R) = \frac{1}{n-1}(H_R - \mu_R)^\top (H_R - \mu_R), \quad (7)$$

where $\mathrm{Cov}(H_N)$ and $\mathrm{Cov}(H_R)$ are the covariance of the node and rule representations, respectively. These matrices provide insights into the variability and relationships between different dimensions of the data. To align the distributions of the node and rule representations, we define the total cross-representation loss, $\mathcal{L}_{\mathrm{cross}}$, as the sum of the mean squared error (MSE) of the mean representations and the MSE of the covariance matrices. This ensures that both the means and standard deviations of the two distributions are matched as follows:

$$\mathrm{MSE}_{\mathrm{mean}} = \frac{1}{d} \sum_{j=1}^{d} \left(\mu_{N,j} - \mu_{R,j}\right)^2,$$
$$\mathrm{MSE}_{\mathrm{cov}} = \frac{1}{d^2} \sum_{j=1}^{d} \sum_{k=1}^{d} \left(\mathrm{Cov}(A)_{jk} - \mathrm{Cov}(B)_{jk}\right)^2, \quad (8)$$

where $\mu_{N,j}$ and $\mu_{R,j}$ are the components of the mean representations $\mu_N$ and $\mu_R$. $\mathrm{Cov}(A)_{jk}$ and $\mathrm{Cov}(B)_{jk}$ are the elements of the covariance matrices of $\mu_N$ and $\mu_R$. $\mathcal{L}_{\mathrm{cross}}$ is then formulated as:

$$\mathcal{L}_{\mathrm{cross}} = \mathrm{MSE}_{\mathrm{mean}} + \mathrm{MSE}_{\mathrm{cov}}, \quad (9)$$

which ensures that the model learns to align the distributions of node representations and rule representations effectively. Ultimately, the comprehensive loss function for Str-GCL is given by:

$$\mathcal{L} = \mathcal{L}_{\mathrm{InfoNCE}} + \mathcal{L}_{\mathrm{rule}} + \mathcal{L}_{\mathrm{cross}}, \quad (10)$$

**Figure 3: The overview of the proposed method.** Two graph views $\mathcal{G}_1$ and $\mathcal{G}_2$ are generated from graph $\mathcal{G}$ by augmentations. NTSC and LGTC process the original graph $\mathcal{G}$ and generate a weight set respectively. Then, each weight is passed to $\text{MLP}_{\text{param}}$ to learn the weights, and finally acts on the representations $H_R$. $\mathcal{G}_1$ and $\mathcal{G}_2$ through a shared GNN encoder generates node representations $U$ and $V$ respectively, and the rule feature $W$ generates the corresponding rule representations $H_R$ through MLP. $H_R$ establishes losses with $U$ and $V$ respectively through $\mathcal{L}_{\text{cross}}$, and constrains nodes to perceive structural commonsense.

where $\mathcal{L}_{\text{InfoNCE}}$ retains the same form as in Equation 1. Overall loss $\mathcal{L}$ integrates the InfoNCE loss, the rule-based loss for incorporating structural commonsense, and the cross-representations loss, providing a mechanism for aligning node and rule representations.

## 5 Related Work

**Graph Contrastive Learning** is currently attracting widespread attention in the academic community. It generates multiple augmented views through data augmentation and designs different objective functions to train the model based on maximizing mutual information, thereby reducing the model's dependence on label information. GRACE [3] trains the model by maximizing the similarity of corresponding nodes in two views and minimizing the similarity between other nodes. On this basis, GCA [10] designs an adaptive enhanced GCL framework to measure the importance of nodes and edges, protecting the semantic information of graph data during augmentation. CCA-SSG [25] utilizes Canonical Correlation Analysis (CCA) [26] to align information from corresponding dimensions across different views while decorrelating information from distinct dimensions, resulting in linear time and space complexity. HomoGCL [12] starts from the assumption of graph homophily and uses a Gaussian mixture model (GMM) to soft-cluster nodes to determine whether neighboring nodes are positive samples. ProGCL [11] uses a Beta Mixture Model (BMM) to estimate the probability that a negative sample is a true negative, and proposes a method to compute the weights of negative samples and synthesize new negative samples. CGKS [27] constructs multi-view GCL models of different scales through graph coarsening and introduces a jointly optimized contrast loss across multiple layers to capture information at different granularities. PiGCL [28] addresses the implicit conflict problem in GCL caused by information mutual exclusion and performs secondary screening of negative samples by dynamically capturing and ignoring conflicting ones.

In BGRL [13], the nodes in the augmented graph are regarded as positive samples, and the online encoder is trained to predict the target encoder to generate efficient node representations. AFGRL [14] differs from augmentation-based GCL methods. It does not rely on data augmentation and negative samples. It discovers positive samples through a $k$-nearest neighbor search and optimizes representation learning by combining local and global information.

Unlike previous models, DGI [2] learns node representations by maximizing the mutual information between node and global representations, treating the corrupted graph as negative samples. GGD [29] designs a new model based on binary cross-entropy loss, analyzing DGI's loss function, and groups positive and negative samples separately, which accelerates the model's training process. Based on DGI, MVGRL [30] generates new structural views through graph diffusion, and distinguishes between the graph representations and node representations generated by different views.

## 6 Experiments

### 6.1 Experimental Setup

We compare Str-GCL with three types of baseline methods, including: (1) Classical unsupervised algorithms: Deepwalk [31] and node2vec [32]. (2) Semi-supervised baselines GCN [23]. (3) GCL baselines: BGRL [13], MVGRL [30], DGI [2], GBT [33], GRACE [3], GCA [10], CCA-SSG [25], Local-GCL [34], ProGCL [11], HomoGCL [12] and PiGCL [28]. We evaluate the effectiveness of Str-GCL using six datasets of different sizes. These datasets include Cora, CiteSeer, PubMed [35], Coauthor CS, Amazon Photo and Amazon Computers [21]. Details are presented in Appendix A.3.

### 6.2 Node Classification

We evaluated the performance of Str-GCL on node classification tasks. During the evaluation phase, we follow the configuration in

**Table 1: Performance on node classification.** $X, A, Y$ denote the node attributes, adjacency matrix, and labels in the datasets. The best and second-best results for each dataset are highlighted in bold and underlined. OOM signifies out-of-memory on 24GB RTX 3090. Data without variance are drawn from previous GCL works[3, 10].

| Method | Available Data | Cora | CiteSeer | PubMed | CS | Photo | Computers |
|---|---|---|---|---|---|---|---|
| Raw Features | $X$ | 64.80 | 64.60 | 84.80 | 90.37 | 78.53 | 73.81 |
| Node2vec | $A$ | 74.80 | 52.30 | 80.30 | 85.08 | 89.67 | 84.39 |
| DeepWalk | $A$ | 75.70 | 50.50 | 80.50 | 84.61 | 89.44 | 85.68 |
| DeepWalk + Features | $X, A$ | 73.10 | 47.60 | 83.70 | 87.70 | 90.05 | 86.28 |
| BGRL | $X, A$ | 81.40 ± 0.57 | 69.53 ± 0.39 | 85.38 ± 0.08 | 92.16 ± 0.13 | 92.75 ± 0.22 | 87.72 ± 0.24 |
| MVGRL | $X, A$ | 84.06 ± 0.63 | 71.78 ± 0.78 | 84.88 ± 0.20 | 92.35 ± 0.14 | 91.94 ± 0.27 | 86.00 ± 0.32 |
| DGI | $X, A$ | 83.71 ± 0.86 | 71.82 ± 1.59 | 86.08 ± 0.23 | 92.87 ± 0.08 | 92.78 ± 0.14 | 87.77 ± 0.36 |
| GBT | $X, A$ | 81.52 ± 0.45 | 68.41 ± 0.66 | 85.81 ± 0.15 | 93.06 ± 0.08 | 92.82 ± 0.40 | 88.85 ± 0.25 |
| GRACE | $X, A$ | 83.96 ± 0.62 | 71.97 ± 0.67 | 86.09 ± 0.17 | 92.19 ± 0.12 | 91.92 ± 0.30 | 88.19 ± 0.41 |
| GCA | $X, A$ | 82.15 ± 1.00 | 69.76 ± 1.05 | 86.58 ± 0.15 | 92.35 ± 0.21 | 91.75 ± 0.29 | 86.58 ± 0.32 |
| CCA-SSG | $X, A$ | 84.06 ± 0.62 | 70.02 ± 1.09 | 86.00 ± 0.22 | 92.05 ± 0.12 | 92.74 ± 0.31 | 88.96 ± 0.13 |
| Local-GCL | $X, A$ | 83.74 ± 0.93 | 70.83 ± 1.62 | 85.89 ± 0.26 | 92.22 ± 0.16 | 92.86 ± 0.23 | 89.54 ± 0.32 |
| ProGCL | $X, A$ | 83.74 ± 0.74 | 71.90 ± 1.66 | 85.84 ± 0.20 | 93.20 ± 0.17 | 92.55 ± 0.38 | 87.69 ± 0.22 |
| HomoGCL | $X, A$ | 83.50 ± 1.09 | 70.34 ± 1.12 | 85.48 ± 0.21 | 91.53 ± 0.13 | 92.35 ± 0.22 | 88.80 ± 0.25 |
| PiGCL | $X, A$ | 84.63 ± 0.78 | 73.51 ± 0.64 | 86.75 ± 0.20 | 93.30 ± 0.09 | 93.14 ± 0.30 | 89.25 ± 0.27 |
| **Str-GCL (Ours)** | $X, A$ | **84.89 ± 0.90** | **73.58 ± 0.84** | **86.81 ± 0.14** | **93.89 ± 0.04** | **93.90 ± 0.26** | **90.19 ± 0.16** |
| Supervised GCN | $X, A, Y$ | 82.80 | 72.00 | 84.80 | 93.03 | 92.42 | 86.51 |

previous works[3, 10], and our GNN encoder and classifier components are the same as those used in GRACE. All of the node classification experiments are shown in Table 1 and our experimental results reveal the following findings: **1)** Our Str-GCL model demonstrated excellent performance across various datasets. In our comparative experiments, our method significantly outperformed the supervised GCN method, underscoring the effectiveness of our approach. **2)** Our model outperforms the baseline model GRACE across all node classification tasks, with significant improvements observed on the CS, Photo and Computers datasets. We analyze the degree and similarity of the datasets and find that there are many high-degree nodes in the CS and Computers. These high-degree nodes make it difficult for local structures to change, and some nodes struggle to break free from the influence of their neighbors solely through the objective function. Structural commonsense enhances and highlights the representations of these nodes during alignment, allowing misclassified nodes to be correctly classified. This corresponds with the earlier results where the proportion of selected rule nodes significantly exceeds the error rate of datasets, indicating that the encoder can indeed learn structural commonsense through rule representations and demonstrating the effectiveness of our rules. In the PubMed dataset, due to its sparsity and generally low node degrees, only the nodes with the smallest degrees are prioritized by structural commonsense. This is to prevent additional information from disrupting the stable structures in the graph. **3)** Our approach significantly outperforms the GRACE-based improved models GCA, Local-GCL, ProGCL, HomoGCL and PiGCL. This further demonstrates that the structural commonsense can indeed enhance model performance.

## 6.3 Ablation Study

In this section, we investigate how each component of Str-GCL, including $\mathcal{L}_{rule}$ and $\mathcal{L}_{cross}$ contributes to the overall performance. The result is shown in Table 2. Here, in "w/o $\mathcal{L}_{rule}$", we disable the $\mathcal{L}_{rule}$ in Equation 5, and in "w/o $\mathcal{L}_{cross}$", we disable the interaction between rule representations and node representations. The ablation study results demonstrate the effectiveness of the proposed loss in our Str-GCL model on different datasets. This trend is consistent across all datasets. Among them, the decrease of deleting $\mathcal{L}_{cross}$ is the most significant compared to only delete $\mathcal{L}_{rule}$, which shows that this alignment mechanism can indeed enable the node representations to perceive the structural commonsense expressed by the rules. In addition, for handling rule representations, $\mathcal{L}_{rule}$ generates a representation space aligned with the node representations, reducing the difficulty of interactions between different representations, which further enhances the performance of the baseline model. When both $\mathcal{L}_{cross}$ and $\mathcal{L}_{rule}$ are eliminated, we observe the most significant decrease, confirming their combined importance in achieving optimal performance.

## 6.4 Performance Analysis of the Str-GCL Plugin

In Table 3, we evaluate the effectiveness of our proposed model by integrating Str-GCL into three classical GCL models: GRACE [3], CCA-SSG [25], and DGI [2]. It is important to note that throughout the paper if Str-GCL is mentioned without specifying a base model, it is implicitly assumed to be based on GRACE for performance evaluation and analysis. During the integration of Str-GCL, we maintain the original parameters of the base models unchanged, modifying only the necessary model architecture and hyperparameters required for the plugin. As shown in the table, the incorporation

Table 2: Ablation study evaluated on six benchmark datasets.

| Model | Cora | CiteSeer | PubMed | CS | Photo | Computers |
|---|---|---|---|---|---|---|
| **Str-GCL** | **84.89 ± 0.90** | **73.58 ± 0.84** | **86.81 ± 0.14** | **93.89 ± 0.04** | **93.90 ± 0.26** | **90.19 ± 0.16** |
| w/o $\mathcal{L}_{\text{rule}}$ | 84.76 ± 0.51 | 73.07 ± 0.31 | 86.62 ± 0.30 | 93.58 ± 0.21 | 93.49 ± 0.41 | 89.88 ± 0.08 |
| w/o $\mathcal{L}_{\text{cross}}$ | 83.73 ± 0.74 | 72.19 ± 1.10 | 86.51 ± 0.15 | 93.74 ± 0.08 | 93.16 ± 0.15 | 89.23 ± 0.13 |
| w/o $\mathcal{L}_{\text{rule}}$ & $\mathcal{L}_{\text{cross}}$ | 83.87 ± 1.10 | 72.23 ± 0.34 | 86.46 ± 0.66 | 93.51 ± 0.15 | 92.95 ± 0.26 | 89.20 ± 0.28 |

Table 3: Node classification accuracy comparison with Str-GCL plugin integration across various GCL models and datasets

| Model | Cora | CiteSeer | PubMed | CS | Photo | Computers |
|---|---|---|---|---|---|---|
| GRACE | $84.0_{\pm0.6}$ | $72.0_{\pm0.7}$ | $86.1_{\pm0.2}$ | $92.2_{\pm0.1}$ | $91.9_{\pm0.3}$ | $88.2_{\pm0.4}$ |
| **Str-GCL$_{\text{GRACE}}$** | $84.9_{\pm0.9}(0.9\uparrow)$ | $73.6_{\pm0.8}(1.6\uparrow)$ | $86.8_{\pm0.1}(0.7\uparrow)$ | $93.9_{\pm0.1}(1.7\uparrow)$ | $93.9_{\pm0.3}(2.0\uparrow)$ | $90.2_{\pm0.2}(2.0\uparrow)$ |
| CCA-SSG | $84.0_{\pm0.6}$ | $70.0_{\pm1.0}$ | $86.0_{\pm0.2}$ | $92.0_{\pm0.1}$ | $92.7_{\pm0.3}$ | $88.9_{\pm0.1}$ |
| **Str-GCL$_{\text{CCA-SSG}}$** | $84.5_{\pm1.1}(0.5\uparrow)$ | $71.3_{\pm0.9}(1.3\uparrow)$ | $86.4_{\pm0.2}(0.4\uparrow)$ | $92.8_{\pm0.1}(0.8\uparrow)$ | $93.2_{\pm0.2}(0.5\uparrow)$ | $89.5_{\pm0.2}(0.6\uparrow)$ |
| DGI | $83.7_{\pm0.8}$ | $71.8_{\pm1.6}$ | $86.0_{\pm0.2}$ | $92.8_{\pm0.1}$ | $92.7_{\pm0.1}$ | $87.8_{\pm0.3}$ |
| **Str-GCL$_{\text{DGI}}$** | $84.4_{\pm0.3}(0.7\uparrow)$ | $72.2_{\pm0.9}(0.4\uparrow)$ | $86.1_{\pm0.3}(0.1\uparrow)$ | $93.3_{\pm0.1}(0.5\uparrow)$ | $93.3_{\pm0.3}(0.6\uparrow)$ | $88.2_{\pm0.2}(0.4\uparrow)$ |

of Str-GCL into various base models leads to performance enhancements across different datasets. Specifically, the performance improvement of Str-GCL$_{\text{GRACE}}$ is the most notable. This is due to the objectives of GRACE, which causes semantically deficient nodes to maintain deficiency while incorporating significant averaging and noises. Consequently, in the InfoNCE loss, the alignment of positive samples lacks learnable information and the discriminative ability between negative samples is diminished. This results in increased bias in the representation space and obscures the core semantics within the embedding space. CCA-SSG employs an invariance loss to align embeddings from different views. Compared to GRACE, CCA-SSG enforces consistency within the representation spaces across different views rather than node-level discrimination. Consequently, CCA-SSG emphasizes the correlation between representations instead of specifically addressing node-level distinctions, resulting in a somewhat reduced performance gain for Str-GCL$_{\text{CCA-SSG}}$ compared to Str-GCL$_{\text{GRACE}}$. DGI maximizes the mutual information between local and global representations, imposing specific constraints on the representation space. However, DGI overlooks the discriminative capacity between nodes, which is a key reason why Str-GCL$_{\text{DGI}}$ can enhance accuracy. Nevertheless, in graphs with high homophily, DGI can still achieve effective representations by solely learning global information.

## 6.5 Error-Prone Nodes Analysis

In this section, we analyze the distribution of misclassified nodes across different datasets using the GRACE model and our proposed Str-GCL model. Table 4 presents the detailed results, showing the number of nodes that were misclassified at least 15 times out of 20 complete runs with Str-GCL. Additionally, it includes the total number of nodes with 15 or more misclassifications. This analysis helps evaluate the effectiveness of Str-GCL in handling frequent errors and identifying unavoidable errors. From Table 4, we can observe that for Str-GCL, the number of frequently misclassified nodes has decreased in each dataset. The most obvious among them

Table 4: Comparsion of misclassified nodes distribution in GRACE and Str-GCL across multiple datasets.

| Datasets | Model | 15 | 16 | 17 | 18 | 19 | 20 | Total | Decline |
|---|---|---|---|---|---|---|---|---|---|
| PubMed | GRACE | 98 | 84 | 93 | 144 | 195 | 1437 | 2051 | - |
| | **Str-GCL** | 100 | 105 | 116 | **138** | 234 | **1254** | **1947** | 5.1% |
| CS | GRACE | 38 | 44 | 40 | 82 | 125 | 804 | 1133 | - |
| | **Str-GCL** | **17** | **31** | **34** | **44** | **65** | **753** | **944** | 16.68% |
| Photo | GRACE | 17 | 16 | 20 | 20 | 29 | 348 | 450 | - |
| | **Str-GCL** | 19 | 17 | 24 | 27 | 35 | **308** | **430** | 4.44% |
| Computers | GRACE | 45 | 36 | 42 | 60 | 73 | 926 | 1182 | - |
| | **Str-GCL** | **35** | **35** | **33** | **55** | 83 | **770** | **1011** | 14.47% |

are the CS and Computers datasets. In the error range of 15-20, the number of almost all misclassified nodes has decreased. This shows that introducing structural commonsense can significantly reduce the number of frequently misclassified nodes. In addition, in the PubMed and CS datasets, although the number of misclassified nodes dropped only 20 times, this also shows that some nodes that will definitely be misclassified can be guided by rules, even if they cannot be completely classified correctly. This will reduce the number of misclassifications of these nodes to a certain extent. This highlights the improved robustness and accuracy of our proposed approach in reducing the most error-prone nodes. Across all analyzed datasets, the Str-GCL model showed clear advantages in handling error-prone nodes. This improvement is attributed to its ability to incorporate structural commonsense that traditional GCL methods such as GRACE cannot capture.

## 6.6 Node Clustering

The experimental results for node clustering are presented in Table 5, where we evaluate the Str-GCL on the Cora, CiteSeer, PubMed and CS datasets. Str-GCL demonstrates excellent performance across multiple clustering tasks. Specifically, Str-GCL outperforms all other

**Table 5: Performance on node clustering. The best and second best results for each dataset are highlighted in bold and underline.**

| Datasets | Model | GRACE | GCA | DGI | BGRL | MVGRL | GBT | Str-GCL (Ours) |
|---|---|---|---|---|---|---|---|---|
| Cora | NMI | 0.5261 | 0.4483 | 0.5310 | 0.4719 | 0.5337 | 0.5055 | **0.5656** |
| | ARI | 0.4312 | 0.3235 | 0.4499 | 0.3851 | 0.4790 | 0.4201 | **0.5067** |
| CiteSeer | NMI | 0.4116 | 0.3909 | 0.3765 | 0.3809 | 0.4133 | **0.4310** | 0.4177 |
| | ARI | 0.4183 | 0.3816 | 0.3752 | 0.3949 | 0.4087 | **0.4400** | 0.4349 |
| PubMed | NMI | **0.3504** | 0.3113 | 0.3128 | 0.2898 | 0.2599 | 0.3266 | 0.3501 |
| | ARI | **0.3307** | 0.3085 | 0.3066 | 0.2645 | 0.2556 | 0.2973 | 0.3069 |
| CS | NMI | 0.7579 | 0.7205 | 0.6062 | 0.6380 | 0.6324 | 0.7524 | **0.7971** |
| | ARI | 0.6538 | 0.5602 | 0.4390 | 0.5346 | 0.5124 | 0.6509 | **0.7852** |

baselines on the Cora and CS datasets and improves on the baseline GRACE by an average of 2.1% in NMI and 5.0% in ARI. Additionally, models based on InfoNCE generally show higher accuracy on the PubMed and CS datasets compared to other GCL baselines, such as BGRL, MVGRL and DGI. However, Str-GCL's accuracy on PubMed does not improve with the incorporation of structural commonsense compared to GRACE. This is attributed to the small difference between inter-class and intra-class similarities in the PubMed dataset, making it difficult to distinguish those nodes at the boundaries of classes.

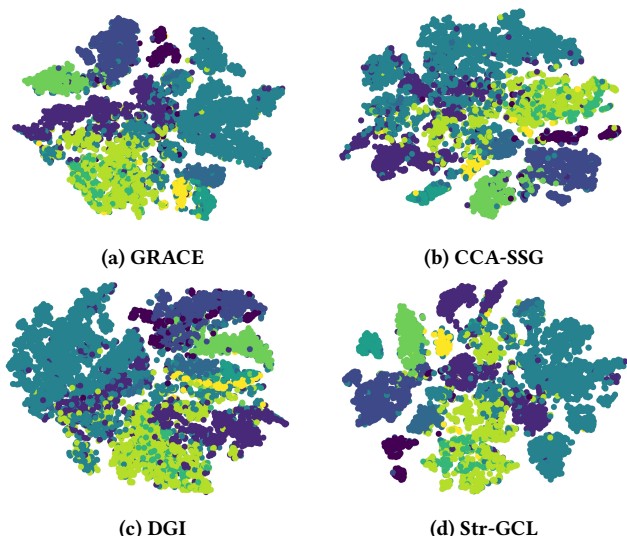

**(a) GRACE**

**(b) CCA-SSG**

**(c) DGI**

**(d) Str-GCL**

**Figure 4: T-SNE embeddings of nodes in Amazon Computers dataset, and the best result is highlighted in underline.**

## 6.7 Visualizaion

In this section, we use T-SNE to demonstrate the advantages of Str-GCL over other baseline models. We conducted experiments on the Amazon Computers dataset using GRACE, CCA-SSG, DGI, and GRACE-Based Str-GCL, as shown in Fig 4. It is evident that, compared to other baseline models, Str-GCL significantly improves the quality of the generated embeddings. While Str-GCL does not further enhance intra-class similarity compared to the default baseline

model GRACE, it optimizes inter-class similarity by increasing the separation between classes and providing stable class assignments for those nodes at the boundaries of classes. This is aligned with the focus of our structural commonsense. DGI, on the other hand, emphasizes local-global similarity, which yields good accuracy in node classification but exhibits less effective clustering. It fails to achieve clear inter-class separation, and the intra-class similarity remains low, particularly in datasets with high homophily. CCA-SSG achieves highly discriminative representations due to the decorrelation between different dimensions. However, in graphs with high homophily, the high similarity between representations increases the difficulty of distinguishing across dimensions, resulting in less effective clustering compared to our method.

## 7 Conclusion

In this paper, we address the limitations of existing GCL methods, which primarily capture implicit semantic relationships but fail to perceive structural commonsense within graph structures. We identify that many nodes with fewer topological connections or lower feature distinctiveness are inadequately trained by conventional GCL methods. To overcome these challenges, we propose a new paradigm called Str-GCL (Structural Commonsense Driven Graph Contrastive Learning), which integrates rules, represented by first-order logic, to guide the model in learning human-perceived structural commonsense, and also provides a new direction for developing universal and efficient rule-based reasoning mechanisms and applying these reasoning rules to existing pre-trained models. Through extensive analysis of various datasets, we demonstrate that manually defined rules can effectively represent structural commonsense from both attribute and topological perspectives. We introduce an alignment mechanism that enables the encoder to perceive these additional structural commonsense, ensuring more comprehensive and effective training for all nodes. We integrate Str-GCL as a plugin into multiple GCL baselines. Extensive experiments and visualization demonstrate the effectiveness of Str-GCL.

Future research could focus on developing automated logic rule definitions to enhance the model's efficiency, scalability, and robustness. Moreover, since our rules operate independently of the encoder, this opens up the possibility of designing a universal rule set and corresponding plugins that can be easily adapted to various models, thereby extending the generalizability of our approach.

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

# A Appendix

## A.1 Proofs for Neighborhood Topological Summation Constraint (NTSC)

**Objection**: Nodes with a smaller sum of neighbor's degrees are poor at average out noise and thus unstable during training.

Let $G = (V, E)$ be an graph where $V$ is the set of nodes and $E$ is the set of edges. For a node $v \in V$, let $N(v)$ denote the set of neighbors of $v$, and let $d(u)$ be the degree of a neighbor $u \in N(v)$. We define the total degree of the neighbors of $v$ as TotalDegree $(v) = \sum_{u \in N(v)} d(u)$. The loss function $L$ can be expressed as the sum of the local losses $L_u$ for each node $u \in V$:

$$L = \sum_{u \in V} L_u, \tag{11}$$

for a sepcific node $v$, the gradient of $L$ with respect to $h_v$ is:

$$\frac{\partial L}{\partial h_v} = \sum_{u \in N(v)} \frac{\partial L_u}{\partial h_v}, \tag{12}$$

each gradient term $\frac{\partial L_u}{\partial h_v}$ may contain a noise component $\epsilon_u$:

$$\frac{\partial L_u}{\partial h_v} = \nabla L_u + \epsilon_u. \tag{13}$$

Thus, the total gradient for node $v$ can be written as:

$$\frac{\partial L}{\partial h_v} = \sum_{u \in N(v)} (\nabla L_u + \epsilon_u) = \sum_{u \in N(v)} \nabla L_u + \sum_{u \in N(v)} \epsilon_u. \tag{14}$$

Let $w$ be a node with a high sum of neighbor degrees TotalDegree$(w)$ and $v$ be a node with a lower sum of neighbor degrees TotalDegree$(v)$, where TotalDegree$(w) >$ TotalDegree$(v)$. Therefore, the noise components for node $w$ and $v$ can be expressed as:

$$\text{Noise }_w = \sum_{u \in N(w)} \epsilon_u, \quad \text{Noise }_v = \sum_{u \in N(v)} \epsilon_u, \tag{15}$$

according to the law of large numbers, as the number of terms increases, the average noise effect decreases:

$$\frac{\text{Noise }_w}{\text{TotalDegree }(w)} \approx \text{E} [\epsilon_u], \quad \frac{\text{Noise }_v}{\text{TotalDegree }(v)} \approx \text{E} [\epsilon_u]. \tag{16}$$

While the expected value of noise $\text{E}[\epsilon_u]$ is the same for all nodes (under the i.i.d. assumption), the actual noise impact on the gradient is smaller for nodes with a higher sum of neighbor degrees due to the averaging effect. Since TotalDegree$(w) >$ TotalDegree$(v)$, the node $w$ with a higher sum of neighbor degrees experiences less relative noise impact:

$$\frac{\text{Noise }_w}{\text{TotalDegree }(w)} < \frac{\text{Noise }_v}{\text{TotalDegree }(v)}. \tag{17}$$

Therefore, nodes with a smaller sum of neighbor degrees are poorer at averaging out noise and have more unstable representations during training.

## A.2 Proofs for Local-Global Threshold Constraint (LGTC)

**Objection**: Nodes with similarity between local and global feature averages have fewer distinctive class features and are more prone to classification errors.

Let $G = (V, E)$ be an undirected graph where $V$ is the set of nodes and $E$ is the set of edges. For a node $v \in V$, let $N(v)$ denote the set of neighbors of $v$. Let $x_v$ represent the original feature vector of node $v$, and $\text{sim}(x_v, x_u)$ is the dot product of the feature between node $v$ and node $u$. Then, we define $\text{LocalSim}(v)$ as the average similarity between $v$ and its neighbor's original features, and define $\text{GlobalSim}(v)$ as the average similarity between $v$ and all other nodes' original features in the graph. The definition is as follows:

$$\text{LocalSim}(v) = \frac{1}{|N(v)|} \sum_{u \in N(v)} \text{sim}(x_v, x_u),$$
$$\text{GlobalSim}(v) = \frac{1}{V} \sum_{u \in V} \text{sim}(x_v, x_u), \tag{18}$$

there we let node $v$ satisfies $|\text{LocalSim}(v) - \text{GlobalSim}(v)|$ is small, and for the representations $h_v^{k+1}$ of node $v$ is as follows:

$$h_v^{(k+1)} = \sigma(\sum_{u \in N(v)} \frac{1}{\sqrt{d(v)d(u)}} W^{(k)} x_u), \tag{19}$$

here we assume $\text{LocalSim}(v) \approx \text{GlobalSim}(v)$, for example, the initial feature of node $v$ and its neighbors is close to the global feature:

$$\text{LocalSim}(v) = \frac{1}{|N(v)|} \sum_{u \in N(v)} (x_v \cdot x_u)$$
$$\approx \text{GlobalSim}(v) = \frac{1}{|V|} \sum_{u \in V} (x_v \cdot x_u), \tag{20}$$

this implies that $x_u$ can be approximately represented by the global feature mean $\overline{x}$:

$$x_u \approx \overline{x} \ \ \forall u \in N(v). \tag{21}$$

Therefore, the updated representation of node $v$ is:

$$h_v^{(k+1)} = \sigma(\sum_{u \in N(v)} \frac{1}{\sqrt{d(v)d(u)}} W^{(k)} \overline{x}). \tag{22}$$

The representation $h_v^{(k+1)}$ of node $v$ primary reflects global features and lacks distinctive class features. Therefore, nodes with similarity between local and global feature averages have fewer distinctive class features and are more prone to classification errors.

## A.3 Datasets

**Table 6: Dataset statistics in experiment**

| Dataset | #Nodes | #Edges | #Features | #Classes |
|---|---|---|---|---|
| Cora | 2,708 | 10,556 | 1,433 | 7 |
| CiteSeer | 3,327 | 9,228 | 3,703 | 6 |
| PubMed | 19,717 | 88,651 | 500 | 3 |
| CS | 18,333 | 163,788 | 6,805 | 15 |
| Photo | 7,650 | 238,163 | 745 | 8 |
| Computers | 13,752 | 491,722 | 767 | 10 |

In Cora, CiteSeer and PubMed[35] dataset, nodes are papers, edges are citation relationships. Each dimension in the feature corresponds to a word. Labels are the categories into which the paper is divided.

Coauthor CS [21] dataset, nodes are authors, that are connected by an edge if they co-authored a paper. Node features represent paper keywords for each author's papers, and class labels indicate most active fields of study for each other.

Amazon Computers and Amazon Photo are segments of Amazon co-purchase graph [36], where nodes represent goods, edges indicate that two goods are frequently bought together, node features are bag-of-words encoded product reviews, and class labels are given by the product category.

## A.4 Pseudo Code of Str-GCL

The following pseudo code outlines the Str-GCL training algorithm, which integrates structural commonsense to enhance GCL. As shown in Algorithm 1. The algorithm identifies error-prone nodes using a set of predefined rules and extracts their original features. During each training epoch, two graph views are generated, and node representations is obtained using an encoder, while rule representations is generated using an MLP. The total loss, comprising contrastive loss, rule loss, and cross loss, is minimized to train the model.

---

**Algorithm 1** The Str-GCL training algorithm

---

**Require:** Original Graph $\mathcal{G}$, Rule Set{ NTSC, LGTC }, Encoder $f$, MLP $g$ and MLP$_{\text{param}}$ $g_{\text{param}}$.

1: Generate weights by applying NTSC and LGTC on the original data
2: Calculate NTSC and LGTC, and get original features $X_R$
3: **for** epoch = 0, 1, 2, . . . **do**
4:   Generate two graph views $\mathcal{G}_1$ and $\mathcal{G}_2$ by corrupting $\mathcal{G}$
5:   Get node representations $U$ of $\mathcal{G}_1$ using the encoder $f$
6:   Get node representations $V$ of $\mathcal{G}_2$ using the encoder $f$
7:   Get rule representations $H_R$ of $X_R$ using the MLP $g$
8:   Get learnable rule weights $w$ and $s$ of $H_R$ using the MLP$_{\text{param}}$ $g_{\text{param}}$
9:   Compute the contrastive loss $\mathcal{L}_{\text{InfoNCE}}$ with Equation 1
10:   Compute the rule loss $\mathcal{L}_{\text{rule}}$ with Equation 5
11:   Compute the cross loss $\mathcal{L}_{\text{cross}}$ to align $U$, $V$, and $H_R$
12:   Update parameters to minimize the total loss $\mathcal{L} = \mathcal{L}_{\text{InfoNCE}} + \mathcal{L}_{\text{rule}} + \mathcal{L}_{\text{cross}}$
13: **end for**
14: **Return** node embedding $H$, trained encoder $f$

---

## A.5 Experimental details

We test Str-GCL on classification and clustering tasks, with both Str-GCL and all GCL baselines trained in a self-supervised manner. For the Cora and CiteSeer datasets, due to their small size, we use a two-layer GCN encoder for training. In contrast, for PubMed, Coauthor CS, Amazon Photo, and Amazon Computers, we employ a single-layer GCN encoder. For the classification task, we follow the same setup as GRACE [3], using 10% of the data to train the downstream classifier and 90% for testing. All experiments are conducted on an RTX 3090 GPU (24GB).

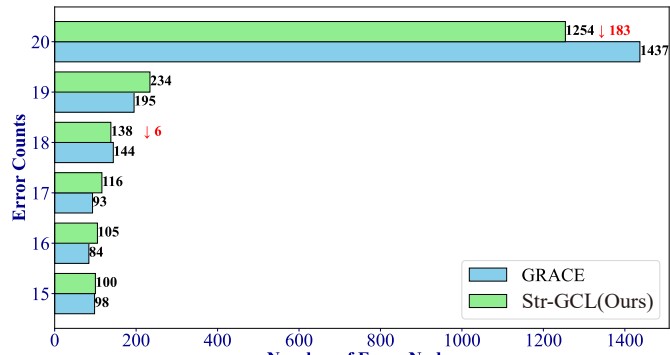

(a) Misclassified nodes distribution comparison of PubMed

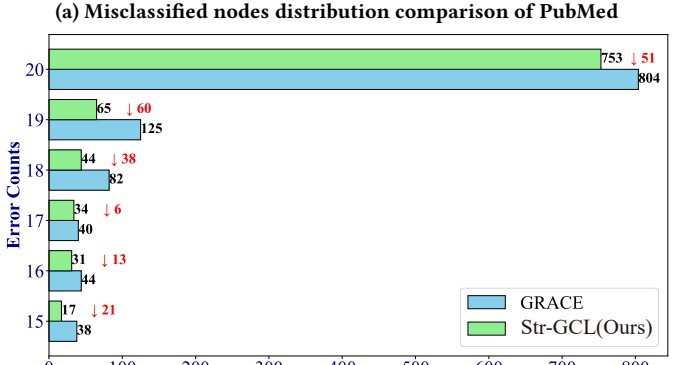

(b) Misclassified nodes distribution comparison of CS

**Figure 5: Misclassified nodes distribution comparison of PubMed and CS datasets.**

## A.6 Hyperparameter Specifications

In this section, we present the hyperparameter specifications used for training the Str-GCL model on various datasets. Table 7 and 8 detail the hyperparameters employed for different datasets.

Table 7 lists the core hyperparameters, including the temperature parameter $\tau$ and $\tau_{\text{rule}}$, learning rate, weight decay, number of epochs, hidden dimension, MLP hidden dimension, and activation function. For example, on the Cora dataset, we used a $\tau$ value of 0.5, a $\tau_{\text{rule}}$ value of 0.4, a learning rate of 0.0001, a weight decay of 0.0005 and 800 epochs, with a hidden dimension of 1024 and an MLP hidden dimension of 128, employing the *relu* activation function.

Table 8 provides additional hyperparameters, such as $\alpha$, $\beta$, drop edge rates, and drop feature rates. For instance, on the Cora dataset, we set drop edge rates of 0.3 and 0.2 for the two dropout layers, and drop feature rates of 0.4 and 0.2. The weight of $\mathcal{L}_{\text{rule}}$ is 100, and the weight of $\mathcal{L}_{\text{cross}}$ is 1.

These hyperparameters are carefully selected to optimize the performance of Str-GCL across different datasets, ensuring robust and consistent results.

**Table 7: Hyperparameters specifications 1.**

| Dataset | $\tau$ | $\tau_{\text{rule}}$ | Learning rate | Weight decay | Num epochs | Hidden dimension | Mlp hidden dim | Activation function |
|---|---|---|---|---|---|---|---|---|
| Cora | 0.5 | 0.4 | 0.0001 | 0.0005 | 800 | 1024 | 128 | *relu* |
| CiteSeer | 0.9 | 0.8 | 0.005 | 0.0001 | 100 | 512 | 256 | *relu* |
| PubMed | 0.9 | 0.9 | 0.0005 | 0.0005 | 1000 | 1024 | 128 | *relu* |
| CS | 0.4 | 0.3 | 0.0005 | 0.00005 | 1000 | 512 | 128 | *relu* |
| Photo | 0.4 | 0.4 | 0.0001 | 0.00001 | 15000 | 2048 | 128 | *relu* |
| Computers | 0.4 | 0.3 | 0.0005 | 0.0001 | 18000 | 512 | 128 | *relu* |

**Table 8: Hyperparameters specifications 2.**

| Dataset | Drop edge rate 1 | Drop edge rate 2 | Drop feature rate 1 | Drop feature rate 2 | Weight of $\mathcal{L}_{\text{rule}}$ | Weight of $\mathcal{L}_{\text{cross}}$ |
|---|---|---|---|---|---|---|
| Cora | 0.3 | 0.2 | 0.4 | 0.2 | 100 | 1 |
| CiteSeer | 0.4 | 0.3 | 0.2 | 0.2 | 5 | 1 |
| PubMed | 0.2 | 0.2 | 0.3 | 0.1 | 1 | 1 |
| CS | 0.1 | 0.2 | 0.3 | 0.1 | 1 | 1 |
| Photo | 0.4 | 0.4 | 0.3 | 0.1 | 1 | 1 |
| Computers | 0.1 | 0.2 | 0.3 | 0.1 | 1 | 1 |

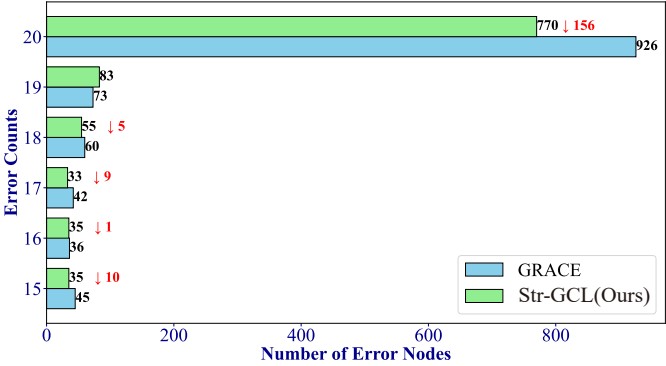

**(a) Misclassified nodes distribution comparison of Computers**

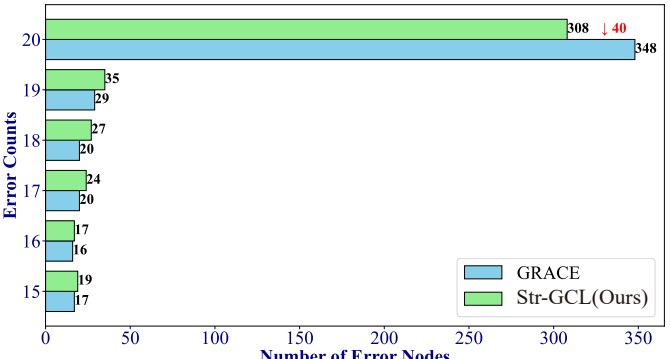

**(b) Misclassified nodes distribution comparison of Photo**

**Figure 6: Misclassified nodes distribution comparison of Computers and Photo datasets.**

## A.7 Misclassified Nodes Analysis on Benchmark Datasets

In this section, we provide a detailed analysis of misclassified nodes across multiple benchmark datasets, following the same experimental settings as described in the main text. As shown in Figure 6 Our goal is to identify nodes that are insufficiently trained, as evidenced by their frequent misclassification errors across multiple tests. As outlined in the main text, we use the well-known GCL method, GRACE [3], and run it 20 times on each dataset under the default experimental settings. For each run, we record the number of misclassifications for each node. This aggregated data allows us to observe the frequency distribution of misclassified nodes and identify those that consistently exhibit high error rates.

The CS dataset demonstrates excellent performance with the Str-GCL model, showing a reduction in the number of misclassified nodes within the 15-20 error range. This indicates that on the CS dataset, Str-GCL not only reduces the number of frequently misclassified nodes but also significantly lowers their error counts. Similarly, The Computers dataset also benefits from the Str-GCL model. Although the reduction in the number of nodes with varying error counts is not as consistent as in the CS dataset, there is a very significant decrease in the number of nodes that are consistently misclassified. This highlights the improved robustness and accuracy of our proposed approach in reducing the most error-prone nodes.

Across all analyzed datasets, the Str-GCL model demonstrates a clear advantage in handling error-prone nodes. This improvement is attributed to its ability to incorporate structural commonsense, which are not captured by traditional GCL methods like GRACE.

## A.8 Reproducibility

Table 9 presents the GitHub links to the source codes of various contrastive methods used in our evaluation.

**Table 9: Code links of various baseline methods.**

| Methods | Source Code |
| --- | --- |
| BGRL | https://github.com/nerdslab/bgrl |
| MVGRL | https://github.com/kavehhassani/mvgrl |
| DGI | https://github.com/PetarV-/DGI |
| GBT | https://github.com/pbielak/graph-barlow-twins |
| GRACE | https://github.com/CRIPAC-DIG/GRACE |
| GCA | https://github.com/CRIPAC-DIG/GCA |
| CCA-SSG | https://github.com/hengruizhang98/CCA-SSG |
| Local-GCL | https://openreview.net/forum?id=dSYkYNNZkV |
| ProGCL | https://github.com/junxia97/ProGCL |
| HomoGCL | https://github.com/wenzhilics/HomoGCL |
| PiGCL | https://github.com/hedongxiao-tju/PiGCL |

