# OpenReview forum: "Str-GCL: Structural Commonsense Driven Graph Contrastive Learning"
_ACM.org/TheWebConf/2025/Conference — WWW 2025 Oral_

### Official Review · Reviewer_T8ak · 2024-11-24

**Novelty:** 3
**Technical Quality:** 4

**Review:**

This paper proposed a method to improve graph contrastive learning method. In particular, it studies the misclassification cases in node classification task trained using classical GCL method and uses "structural commonsense" to account for the misclassified cases. It uses the the rules derived from the structural commonsense to design a rule-based embedding, and in the end, loss functions that align the rule-based embedding space and the GNN encoder's embedding to generate better representations to address the misclassified nodes. It ends up demonstrating some marginal improvements over existing models on node classification tasks.

The paper, while showing some improvements on the experiments, has several major weaknesses that it needs to address:
1. __The lack of a global analysis on nodes before and after apply their plugin__: in the beginning they studied the "frequently misclassified nodes" and showed that in the end the error rates drop; however, their overall performance increase is not exactly equal to the error rate drop, so what happened to _other nodes that are not in this category of frequently misclassified nodes_? Does adding this plugin deteriorate performance for some other nodes?
2. __The sole focus of using GCN as the graph encoder__: they only performed case study on GCN, which is the most basic graph encoder; however, if a more powerful GNN encoder, such as those that leverage topological information, were used in the study, would the plugin's influence be meaningful? This needs to be studied or at least be mentioned in their research outlook.
3. __The lack of clarity in writing (I)__: There are two major weaknesses in the author's writing; the first weakness is the lack of clear definition of their main concept, "structural commonsense". Their illustrated examples in 4.1 is awkward in that it is neither rigorous nor concrete enough. It would be helpful to have a section where it starts with something like __Definition (Structural Commonsense)__ and provide a definition for readers to better grasp this idea. Or they should clearly state it in the abstract or the introduction section, rather than just using this word and assume readers are all familiar with what it entails.
4. __The lack of clarity in writing (II)__: the mathematical symbols are sometimes thrown out in the paper without clear definitions; an example is in L378, where the symbols $q$ and $s$ were used, but surprisingly these two never showed up again in their loss calculations. When the paper is full of different quantities, it is advised that the authors provide a symbol reference table for readers to follow on what each letters represent.
5. __The lack of fundamental connection between their problem of the use of First Order Logic__: by using FOL, their motivation seems to be drawing connection between it and the notion of commonsense, I draw this conclusion since their loss design end up has little to do with FOL (at least directly). This could be an artifact of their writing, as the logic part seems to be unimportant in general and more like a side note or an addendum.

6. __The problem of information leaking__: in equation 6 and 7, the statistics were calculated using "error-prone nodes". However, these nodes can only be obtained after you run the entire evaluation. But SSL should be something performed beforehand, so how do you get these nodes in a graph that doesn't have the evaluation results available? This seems to be "training using ground truth from test set" scenario.

Overall, despite some improvements on the experimental results and some insights provided, I think the paper requires many improvements.

**Questions:**

As indicated in the review section, there are several questions:
1. what exactly is the definition of "structural commonsense"?
2. how does the other nodes behave after adding your plugin?
3. what are the meaning of the mathematical symbols?(symbol table)
4. how does the framework fare with more advanced GNN models?
5. Does FOL matter for your loss design? the connection is not clear here.
6. How do you address point 6 in the comment?

**Reviewer Confidence:**

3: The reviewer is confident but not certain that the evaluation is correct

**Scope:**

4: The work is relevant to the Web and to the track, and is of broad interest to the community

---

### Official Review · Reviewer_Qb1w · 2024-11-25

**Novelty:** 5
**Technical Quality:** 6

**Review:**

This paper proposes a novel graph contrastive learning framework Str-GCL that leverages first-order symbolic logic rules to represent structural commonsense and explicitly integrates these rules into the GCL framework. Experiments demonstrate that Str-GCL outperforms existing GCL methods, providing a new perspective on leveraging structural commonsense in graph representation learning.

## Pros:

1.	The authors conducted comprehensive experiments across multiple datasets, demonstrating the robustness and versatility of their proposed method in diverse scenarios.
2.	The paper introduces a novel heuristic approach that aims to address the issue of error-prone nodes, enhancing the model's performance in downstream tasks.
3.	The paper is easy to follow and well-organized.

## Cons:

1.	The paper relies on two key assumptions: that higher-degree nodes are more reliable and that nodes with similar local and global similarities are harder to classify, termed structural commonsense. However, the paper does not provide adequate references to support these assumptions as widely accepted in the community, nor sufficient experimental evidence to validate their correctness.
2.	Different illustrations about structural commonsense. In Figure 1, the similarity is calculated using node features and global features, whereas in Section 4.1, it is computed using the mean neighbor features and global features. This inconsistency creates confusion and poses an obstacle to understanding the concept of structural commonsense.
3.	In Table 2, even when excluding $ \mathcal{L}\_{rule} $ and  $ \mathcal{L}\_{cross}$, which causes Str-GCL to revert to its base model, it still achieves a better performance (93.51% accuracy) compared to the best baseline (93.30%) on the CS dataset.

**Questions:**

- Are the two rules: NTSC and LGTC harder to classify, and widely accepted by the community? Is there evidence to prove their general applicability and reliability?
- Why does Str-GCL still perform better than baselines without $\mathcal{L}\_{rule}$ and $\mathcal{L}\_{cross}$?

**Reviewer Confidence:**

3: The reviewer is confident but not certain that the evaluation is correct

**Scope:**

4: The work is relevant to the Web and to the track, and is of broad interest to the community

---

### Official Review · Reviewer_TcbM · 2024-12-02

**Novelty:** 5
**Technical Quality:** 6

**Review:**

## Summary

This manuscript introduces structural commonsense from both topological and attribute rule perspectives and designs a representation alignment mechanism to effectively capture structural commonsense. Experimental results demonstrate that the proposed Str-GCL outperforms existing GCL methods in representation learning and clustering.

---

## Pros

1. The motivation of the idea is interesting. The analysis based on the error ratio is insightful.
2. The proposed method achieves excellent performance on node classification tasks. And as a plug-and-play approach, it can be well integrated with various graph contrastive learning methods.

---

## Cons

1. In Table 1, the authors illustrate the performance of supervised GCN. However, based on recent work [1], classic GNNs (e.g., GCN) are strong node classifiers. I understand that comparing supervised GCN with a linear probe is unfair, but to avoid misleading readers, it is better to replace it with the latest results.
2. The authors only conduct experiments on node-level representation learning scenarios, and it lack experiments on graph-level tasks. In my opinion, graph-level pretraining is much more important than node-level pretraining.
3. Since the proposed Str-GCl is a plug-and-play module, providing complexity analysis is necessary.

**Questions:**

Questions: See Cons in Review.

**Reviewer Confidence:**

3: The reviewer is confident but not certain that the evaluation is correct

**Scope:**

4: The work is relevant to the Web and to the track, and is of broad interest to the community

---

### Official Review · Reviewer_rGk2 · 2024-12-02

**Novelty:** 5
**Technical Quality:** 5

**Review:**

This paper proposes a Structural Commonsense Unveiling in Graph Contrastive Learning (Str-GCL) framework to identify and integrate structural commonsense in graph contrastive learning. This framework introduces structural commonsense from both topological and attribute rule perspectives and design a representation alignment mechanism to guide the encoder to capture this structural commonsense.

Strengths:

1. The paper is well-motivated, with a clear  rationale for addressing the misclassification problem in current GCL methods.

2. This paper proposes a framework to integrate structural commonsense into graph contrastive learning.


Weaknesses:

1. The description of the proposed method lacks clarity, particularly in terms of the generation process of the rule representation $H_R$ . Additionally, lines 376–381 are ambiguous: it is unclear what is $q$ and what relationship of $q$ and $H_R$? And what is the features $W$?

2. The role of first-order logic in the proposed method is not clearly described. Moreover, the lack of a definition or explicit use of first-order logic does not appear to impact the implementation of the proposed method.

**Questions:**

1. The description of the proposed method lacks clarity, particularly in terms of the generation process of the rule representation $H_R$ . Additionally, lines 376–381 are ambiguous: it is unclear what is $q$ and what relationship of $q$ and $H_R$? And what is the features $W$?

2. The role of first-order logic in the proposed method is not clearly described. Moreover, the lack of a definition or explicit use of first-order logic does not appear to impact the implementation of the proposed method.

**Reviewer Confidence:**

3: The reviewer is confident but not certain that the evaluation is correct

**Scope:**

4: The work is relevant to the Web and to the track, and is of broad interest to the community

---

### Official Review · Reviewer_JmKM · 2024-12-02

**Novelty:** 5
**Technical Quality:** 5

**Review:**

The paper introduces Str-GCL, a graph contrastive learning model that integrates structural commonsense, represented as first-order logic rules, to enhance the learning process and address the limitations of existing GCL methods in perceiving complex structural patterns within graph data. Str-GCL incorporates two rules, NTSC and LGTC, which respectively focus on nodes with limited topological connections and those exhibiting significant differences between local and global feature similarity. Through a representation alignment mechanism and a comprehensive loss function, Str-GCL effectively guides the encoder to learn from both the explicit structural commonsense encoded in the rules and the implicit relationships captured by the GCL framework, leading to improved performance in node classification and clustering tasks.

pros:
1.  The paper identifies a significant limitation in existing GCL methods: the inability to effectively capture and utilize structural commonsense present in graph data. By recognizing and explicitly addressing this gap, this paper introduces a novel and insightful research direction in GCL.
2. This paper proposes Str-GCL, which is a novel GCL framework and directly incorporates human-understandable structural commonsense into the learning process using first-order logic rules.
3. This paper provides a comprehensive empirical evaluation of Str-GCL on six datasets, demonstrating its effectiveness in both node classification and clustering tasks.

cons:
1. This paper focuses on two specific rules, NTSC and LGTC, to represent structural commonsense. While these rules are well-motivated and demonstrate effectiveness in addressing certain types of node misclassifications, they may not be comprehensive enough to capture the full range of structural commonsense relevant to different graph datasets and tasks.
2. The addition of rule-based representations and the alignment mechanism introduces additional computational complexity compared to traditional GCL methods. An analysis of the computational overhead associated with Str-GCL would further strengthen the paper.

**Questions:**

Please refer to cons.

**Reviewer Confidence:**

3: The reviewer is confident but not certain that the evaluation is correct

**Scope:**

4: The work is relevant to the Web and to the track, and is of broad interest to the community